# Self-Reported Depression among Chinese Women with Recurrent Pregnancy Loss: Focusing on Associated Risk Factors

**DOI:** 10.3390/jcm11247474

**Published:** 2022-12-16

**Authors:** Rui Gao, Lang Qin, Peng Bai

**Affiliations:** 1The Reproductive Medical Center, Department of Obstetrics and Gynecology, West China Second University Hospital, Sichuan University, Chengdu 610041, China; 2Key Laboratory of Birth Defects and Related Diseases of Women and Children of the Ministry of Education, West China Second University Hospital, Sichuan University, Chengdu 610041, China; 3West China School of Medicine, Sichuan University, Chengdu 610041, China; 4Department of Forensic Genetics, West China School of Basic Medical Sciences & Forensic Medicine, Sichuan University, Chengdu 610041, China

**Keywords:** depression, recurrent pregnancy loss, risk factors, prevalence, RPL, cross-sectional study

## Abstract

(1) Background: To investigate whether women suffering from recurrent pregnancy loss (RPL) have a higher prevalence of self-reported depression than healthy controls and to assess the associated risk factors for RPL women self-reporting the symptom of depression. (2) Methods: A cross-sectional study investigating 247 women with histories of RPL and 193 healthy women was performed in Southwest China. The Zung Self-Rating Depression Scale (SDS) was used to measure self-reported depression, and the prevalence of self-reported depression was compared between the two groups. Sociodemographic data for the two groups and clinical information for the RPL group were collected by questionnaires. (3) Results: The prevalence of self-reported depression was higher in the RPL group than in the control group (45.3% vs. 30.1%, *p* < 0.01). Subgroup analyses indicated that the statistical difference in the prevalence of self-reported depression was significant in the subgroups of women in the first trimester of gestation, age ≥ 36 years, BMI ≥ 18.5 kg/m^2^, working hours ≤ 8 h/day, university and higher education, and urban residence. Multivariable logistic analysis indicated that age ≥ 36 years, >2 times of spontaneous miscarriages, and no history of live birth were independent risk factors for RPL women self-reporting depression. (4) Conclusions: A higher prevalence of self-reported depression was observed in RPL patients than in healthy women. The psychological status for RPL patients with age ≥ 36 years, >2 times of spontaneous miscarriages, or without a history of a live birth needs to be further addressed.

## 1. Introduction

Recurrent pregnancy loss (RPL) is defined as two or more consecutive pregnancy losses before 24 weeks of gestation [1]. It was reported that about 1% to 5% of childbearing women have been affected by RPL [2]. The etiologies of RPL include endocrinological disorders, chromosome abnormalities, anatomical abnormalities, immune disorders, pre-thrombotic state, infection, and other factors [1,3], but in nearly 25% of cases, a specific etiology cannot be determined [4]. As reported, depression is a common negative emotion that has been found to be associated with RPL [5,6,7]. A nested case-control study involving 2558 participants indicated that depression has a synergistic effect after the first pregnancy loss, which increases the incidence of subsequent RPL [7]. The Practice Committee of the American Society for Reproductive Medicine (ASRM) lists psychological factors as independent factors of RPL and recommends offering these patients psychological support and counseling [3].

A few previous studies explored the effects of RPL on psychological disorders such as depression [5,8,9,10,11]. However, variable complex psychological scales limited the generalization of these findings, and we still do not know the risk factors of self-reported depression in patients with RPL, which is crucial for clinicians to identify risky persons and perform early psychological intervention. The self-rating depression scale (SDS) is a simple and reliable tool for assessing depressive symptoms, and it is widely applied in clinical and epidemiological studies [12]. This scale contains 20 items that describe subjective feelings and the manifestation of depression, which were all straightforward and self-reported by the informant. It is user-friendly and therefore, suitable for clinician use in decide which patients need more psychological care.

For the above reasons, we performed this cross-sectional study in Southwest China to investigate whether women suffering from RPL have higher prevalence of self-reported depression than healthy controls, and to explore the associated risk factors for RPL women self-reporting the symptom of depression using the SDS scale.

## 2. Materials and Methods

### 2.1. Study Design

A cross-sectional study was performed at West China Second University Hospital, Chengdu, Sichuan Province, from April 2021 to February 2022. Medical records from reproductive centers, obstetrics centers, and health examination centers were examined, among which 250 women with a history of RPL and 200 healthy women were invited to participate in this program. In the end, 247 women with a history of RPL, 31 healthy non-pregnant women, and 162 healthy pregnant women agreed to participate in the project. A QR code for an electronic questionnaire was sent to all participants; under staff supervision, they scanned this QR code and completed the questionnaire independently using smartphones. After these questionaries were submitted, the results were exported to a previously designed form. The ethics committee of West China Second University Hospital approved this study. All participants provided informed consent prior to the study’s commencement. The results of this study were reported according to the STROBE statement.

### 2.2. Participants

In this study, 247 women with a history of RPL were included in the RPL group. The diagnostic criteria for RPL were two or more consecutive failed pregnancies before 24 weeks of gestation, only including intrauterine pregnancy loss confirmed by a transvaginal ultrasound examination. Ultrasound examination results were also required to satisfy at least one of the following [13]: (a) no fetal heartbeat was observed, with a head-to-hip diameter ≥ 7 mm, (b) no embryos were observed, with a mean gestational sac diameter ≥ 25 mm, (c) pregnancy sacs without yolk sacs showed no embryo or fetal heartbeat after two weeks, (d) the pregnant sac with the yolk sac did not show an embryo or fetal heartbeat after 11 days. Subjects with diagnoses of depression and other psychological problems, or who were currently using psychotropic drugs, were excluded.

A total of 193 healthy women were selected as controls in this study. The inclusion of a control group was based on the following items: (a) no previous history of pregnancy and not diagnosed with infertility; (b) no history of assisted reproductive treatment; (c) no history of irregular menstruation or other gynecological disease; (d) free from chronic diseases, such as liver disease or endocrine disease, (e) no family history of genetic diseases, (f) no history of hypertension (a systolic blood pressure ≥ 140 mm Hg or a diastolic blood pressure ≥ 90 mm Hg), (g) no abnormalities in the heart, liver, lungs, or kidneys, (h) no surgery within the previous four months, and (i) no drugs intake or antibiotics misuse within two weeks prior to the study. All subjects were women between 20 and 50 years old, and no psychological interventions were performed before joining the investigation.

### 2.3. Questionnaire

A custom-made questionnaire was used to collect sociodemographic data and the SDS scores of all participants, as well as clinical information for the women with RPL (as shown in Appendix A). Sociodemographic data from the RPL group and the control group consisted of gestational status (non-pregnancy, first, second, or third trimester), age (≤35 years or ≥36 years), body mass index (BMI) (<18.5 or 18.5–24 or >24 kg/m^2^), educational background (university and higher or lower levels of education), working hours (≤8 or >8 h/day), smoking (yes or no), alcohol consumption (yes or no), and residence (rural or urban). Household income (≤CNY 10,000 or >CNY 10,000/month) was investigated only in the RPL group because a few non-pregnant women in the control group were unmarried. Clinical information from the RPL groups consisted of times of spontaneous miscarriages (2 or >2), history of stillbirth (yes or no), history of induced abortion (yes or no), and history of live birth (yes or no). All of the information collected above were determined according to a previous article exploring the association between depression and RPL, as well as clinical experiences [5,7,9,11]. In addition, participants had to rate the 20 items of SDS on a 1–4 scale (1 = never or rarely, 2 = sometimes, 3 = often, and 4 = most of the time), the sum of the scores for all items is the crude SDS score, and the total SDS scores were defined as crude SDS scores * 1.25 (reserve integer). A total SDS score below 53 was regarded as not symptomatic of depression, according to the cut-off values established for Chinese patients [14].

### 2.4. Statistical Analysis

Missing data in this study was filled via the mean filling method (if missing data accounted for less than 5% of the total). Categorical variables were expressed as percentages and were compared via the chi-squared test; if numbers were less than 5 in at least 20% of the cells, Fisher’s exact test was performed. The risk factors for depression in the RPL group were first detected by univariable logistic regression analysis, and significant risk factors confirmed by univariable logistic regression analysis were included in a multivariable logistic regression analysis. Statistical analyses were performed using SPSS, version 25.0 (IBM Corp., Armonk, NY, USA), and the forest plot for subgroup analysis was performed using the “forestplot” package (version 3.1.0) in RStudio, version 3.4.3; *p*-values less than 0.05 were considered statistically significant.

## 3. Results

### 3.1. Sociodemographic Characteristics between RPL Group and Control Group

The sociodemographic characteristics of the participants in the RPL group and the control group are summarized in Table 1. Among the 247 women in the RPL group, 72 were not pregnant, 106 were in the first trimester of gestation, and 69 were in the second or third trimester of gestation. A total of 155 patients had experienced two spontaneous miscarriages, and 92 had experienced three or more spontaneous miscarriages. A total of 35 women had a history of stillbirth, and 166 patients had a history of induced abortion. There were statistical differences in the gestational status, age, educational background, and residence between the two groups (chi-squared test; all *p* < 0.01), but the proportion of BMI, working hours, smoking, and alcohol consumption were not statistically different between the two groups. The prevalence of self-reported depression in the RPL group was significantly higher than that in control group (45.3% vs. 30.1%; chi-squared test, *p* < 0.01).

### 3.2. Subgroup Analyses of Self-Reported Depression between RPL Group and Control Group

We performed subgroup analyses according to gestational status, age, BMI, working hours, educational background and residence to explore the influence of these parameters on the results (as shown in Figure 1); smoking and alcohol consumption were not included in the subgroup analyses because of the small sample size in the subgroup. The results of univariable regression analysis indicated that the prevalence of self-reported depression was higher in the RPL group than in the control group for patients in the first trimester of the gestation subgroup [RR = 3.05 (1.56–5.96), *p* < 0.01], but the difference in the non-pregnant, second, or third trimester subgroup was not significant [RR = 1.85 (0.75–4.57) and 1.27 (0.67–2.41); *p* = 0.183 and 0.473]. In both age subgroups, the prevalence of self-reported depression was higher in the RPL group than in the control group. For women with BMI 18.5–24 kg/m^2^ or >24 kg/m^2^, the prevalence of self-reported depression was higher in the RPL group than in the control group [RR = 1.71 (1.06–2.76) and 3.00 (1.26–7.13); *p* = 0.027 and 0.013], but this difference was not observed for women with BMI < 18.5 kg/m^2^. For women in the working hours ≤ 8 h/day subgroup, the university and higher education subgroup, and the urban subgroup, we also found a higher prevalence of self-reported depression in the RPL group than in the control group.

### 3.3. Risk Factors for Self-Reported Depression among RPL Patients

To explore the risk factors for self-reported depression in the RPL patients, we divided all RPL patients into depression group and no depression group (as shown in Table 2). There were significant differences in age, education background, times of spontaneous miscarriages, and history of live birth between the two groups. By univariable regression analysis, the depression group showed higher proportions of age ≥ 36 years (26.8% vs. 8.1%, *p* < 0.01), lower levels of education (50.0% vs. 35.6%, *p* = 0.02), >2 times of spontaneous miscarriages (50.0% vs. 26.7%, *p* < 0.01) and no live birth (86.6% vs. 68.1%, *p* < 0.01). The above different risk factors were all included in the multivariable logistic regression analysis, and the results showed that age ≥ 36 years [RR = 5.47 (2.42–12.38); *p* < 0.01], ≥2 times of spontaneous miscarriages [RR = 2.94 (1.66–5.22); *p* < 0.01], and no live birth [RR = 3.77 (1.81–7.82); *p* < 0.01] were independent risk factors of self-reported depression for RPL patients (as shown in Table 3).

## 4. Discussion

According to previous studies, the prevalence of RPL in the general population was 1.8% to 2.6% [4,15], and the incidence of RPL has been increasing in recent years. Depression is a common psychological disorder and was regarded as a potential risk factor of many adverse pregnancy outcomes or pregnancy complications, such as RPL [3,16,17]. According to a Chinese investigation in 2019, the lifetime prevalence of depression among the Chinese population was 6.9% [18]. Previous studies indicated that RPL increased the incidence of depression compared with the risk in healthy controls (OR = 3.88; 95% CI = 1.87–8.03) [15]. An increased incidence of post-traumatic stress disorder and suicide have also been observed in RPL patients [15]. Likewise, psychological intervention has proven to improve the pregnancy outcomes of RPL patients. Psychological management for RPL patients was suggested by a small prospective study of 45 pregnancies with prior history of RPL, with other causes eliminated. The patients in this study completed a group of self-reporting questionnaires and treatments before their subsequent pregnancies. A total of 10 of the pregnancies (22.2%) resulted in a miscarriage, which significantly, was predicted by the degree of baseline depression symptoms [19]. A cohort study performed routine obstetrical care and tender-loving care for 42 and 116 RPL patients, respectively. The results showed that RPL patients who received tender-loving care had a significantly higher live birth rate than those who received routine obstetrical care (85% vs. 36%). Two other non-randomized studies also showed a significant improvement in subsequent pregnancy outcomes when close monitoring and support at a dedicated RPL clinic was provided [20,21]. It seems that psychological intervention should be performed in all RPL patients. However, psychological support is expensive and difficult to carry out in some regions. In addition, these estimating tools used in previous studies restricted the clinical performance and were not sensitive for RPL patients with depression symptoms. In addition, early identification of RPL patients with risk factors for depression, as well as psychological intervention, may allow for a more economical use of resources, so it is important to find common risk factors of depression for RPL patients.

In this study, SDS was used to evaluate the self-reported depressive symptoms of the participants. In total, the prevalence of self-reported depression was higher in RPL patients than in healthy women, especially for women in the first trimester of gestation, over 35 years of age, with a BMI equal or more than 18.5 kg/m^2^, working ≤ 8 h/day, with a university or higher education, and living in an urban environment. By multivariable logistic analysis, we found that age ≥ 36, >2 times of spontaneous miscarriages, and no history of live birth were independent risk factors of self-reported depression for RPL patients. These results were consistent with those of previous studies. A study performed by Toffol et al. found that a higher number of miscarriages was associated with a worse current state of mood and a higher frequency of a psychiatric disorder [22]. He L et al. also confirmed that women with ≥3 pregnancy losses were significantly more depressed than women with 2 pregnancy losses [5]. In our study, >2 times of spontaneous miscarriages was proved as an independent risk factor of self-reported depression for RPL patients. We concluded that a higher number of miscarriages may be a trigger that aggravates the psychological stress of the patients. A history of no live births was also significantly associated with depression, consistent with the results of previous studies [23,24,25], which demonstrated that women who were involuntarily childless were more likely to be psychologically distressed, with complicated grief and poor perceived social support. Furthermore, we found that age ≥ 36 years was another important and independent risk factor of self-reported depression in RPL patients. It may be explained by the fact that older RPL patients were under more social pressure. We tried to explore lower levels of education as risk factors of self-reported depression, which was confirmed by He L et al. [5]., but we found lower levels of education were not an independent risk factor in this study. We did not investigate the effect of marital relationship and male factors on self-reported depression of the RPL women, but we find it interesting to explore whether and how the marital relationship and husband may affect the psychologic status of women with RPL.

We must realize that the SDS only provides information regarding a primary estimation of depression symptoms by self-reporting, which differs from clinically diagnosed depression. Thus, this can explain why the prevalence of self-reported depression was so high in both the RPL and control groups; another explanation is that the prevalence of clinical depression is higher in pregnant women than in non-pregnant women [17,26,27]. This study has many implications for us, and we should pay full attention to the psychological status of RPL patients, especially for RPL patients with these three risk factors. SDS is a sensitive and simple tool for clinicians to use to assess the depressive symptoms of RPL patients; for patients with self-reported depression, we can refer them to a psychologist, which is beneficial for both physical and mental health, and can even help to improve the success rate of subsequent pregnancies.

Although this study has important implications for the clinical management of patients with RPL, the results should be interpreted with caution due to its various limitations. Firstly, this study used SDS as scales for evaluating depression symptoms because patients find them easy to understand, but they are not professional diagnostic tools for depression and can only be used for an initial screening [12]. Secondly, there were statistical differences in gestational status, age, educational background, and residence between the RPL group and the control group, which may reduce the accuracy of the results indicating that the RPL patients have a higher prevalence of self-reported depression. Therefore, we must interpret this result with caution. Thirdly, both RPL patients and healthy women in this study were selected from a university hospital women’s medical center in Southwest China. Although we set strict inclusion criteria to exclude controls with physical disease, the medical visit process may have also produced adverse psychological implications for these participants, which could have led to an overestimation of prevalence. In addition, this study followed a single-center design, and the sample size in this study was limited. Instead of dynamically observing the anxiety and depression scores of RPL patients throughout their subsequent pregnancies, we only investigated their anxiety and depression at certain time points. Such sampling cannot explain the causal relationship that RPL has with anxiety and depression. In the future, multicenter studies with large sample sizes and community controls will be necessary to verify the results of this study.

## 5. Conclusions

In conclusion, a higher prevalence of self-reported depression was observed in RPL patients than in healthy women. Clinicians should pay more attention to the depressive symptoms of RPL patients, especially those whose age is equal to or greater than 36 years, who have experienced more than two spontaneous miscarriages, and who have no history of a live birth.

## Figures and Tables

**Figure 1 jcm-11-07474-f001:**
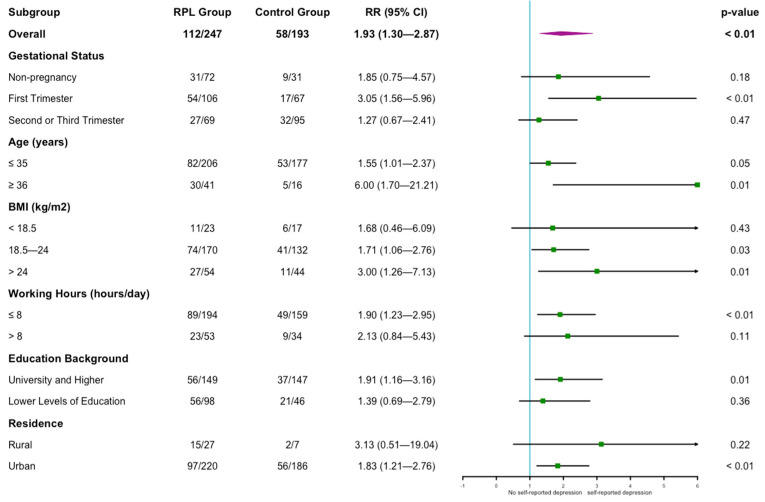
Forest plot of subgroup analyses of self-reported depression between the two groups. RR: relative risk; CI: confidential intervals. *p*-values were calculated by univariable regression analysis, and *p* < 0.05 was regarded as a statistical difference.

**Table 1 jcm-11-07474-t001:** The Sociodemographic and clinical information of the RPL group and the control group.

Parameters	RPL Group (*n* = 247)		Control Group (*n* = 193)		*p*-Value
	*n*	%	*n*	%	
Gestational Status					
Non-pregnant	72	29.2%	31	16.1%	<0.01
First Trimester	106	42.9%	67	34.7%	
Second or Third Trimester	69	27.9%	95	49.2%	
Age (years)					
≤35	206	83.4%	177	91.7%	<0.01
≥36	41	16.6%	16	8.3%	
BMI (kg/m^2^)					
<18.5	23	9.3%	17	8.8%	0.96
18.5–24	170	68.8%	132	68.4%	
>24	54	21.9%	44	22.8%	
Working Hours (hours/day)					
≤8	194	78.5%	159	82.4%	0.19
>8	53	21.5%	34	17.6%	
Education Background					
University and Higher	149	60.3%	147	76.2%	<0.01
Lower Levels of Education	98	39.7%	46	23.8%	
Smoking					
Yes	7	2.8%	9	4.7%	0.22
No	240	97.2%	184	95.3%	
Alcohol Consumption					
Yes	3	1.2%	6	3.1%	0.15
No	244	98.8%	187	96.9%	
Residence					
Rural	27	10.9%	7	3.6%	<0.01
Urban	220	89.1%	186	96.4%	
Times of Spontaneous Miscarriage					
2	155	62.8%	—	—	—
>2	92	37.2%	—	—	
History of Stillbirth					
Yes	35	14.2%	—	—	—
No	212	85.8%	—	—	
History of Induced Abortion					
Yes	166	67.2%	—	—	—
No	81	32.8%	—	—	
History of Live Birth					
Yes	28	11.3%	—	—	—
No	219	88.7%	—	—	
Household Income (CNY/month)					
≤10,000	92	37.2%	—	—	—
>10,000	155	62.8%	—	—	
Self-Reported Depression	112	45.3%	58	30.1%	<0.01

Clinical information regarding times of spontaneous pregnancy loss, history of pregnancy loss > 12 weeks, history of induced abortion, live birth, and household income were not investigated in the control group. BMI: body mass index. *p*-values were calculated by chi-squared test, and *p* < 0.05 was regarded as a statistical difference.

**Table 2 jcm-11-07474-t002:** Univariable analysis of risk factors of self-reported depression for RPL patients.

Parameters	Depression Group (*n* = 112)		No Depression Group (*n* = 135)		*p*-Value
	n	%	n	%	
Gestational Status					
Non-pregnant	31	27.7%	41	30.4%	0.28
First Trimester	54	48.2%	52	38.5%	
Second or Third Trimester	27	24.1%	42	31.1%	
Age (years)					
≤35	82	73.2%	124	91.9%	<0.01
≥36	30	26.8%	11	8.1%	
BMI (kg/m^2^)					
<18.5	11	9.8%	12	8.9%	0.69
18.5–24	74	66.1%	96	71.1%	
>24	27	24.1%	27	20.0%	
Working Hours (hours/day)					
≤8	89	79.5%	105	77.8%	0.75
>8	23	20.5%	30	22.2%	
Education Background					
University and Higher	56	50.0%	87	64.4%	0.02
Lower Levels of Education	56	50.0%	48	35.6%	
Smoking					
Yes	6	5.4%	1	0.7%	0.07
No	106	94.6%	134	99.3%	
Alcohol Consumption					
Yes	3	2.7%	0	0.0%	0.06
No	109	97.3%	135	100.0%	
Residence					
Rural	15	13.4%	12	8.9%	0.26
Urban	97	86.6%	123	91.1%	
Times of Spontaneous Miscarriages					
2	56	50.0%	99	73.3%	<0.01
>2	56	50.0%	36	26.7%	
History of Stillbirth					
Yes	17	15.2%	18	13.3%	0.68
No	95	84.8%	117	86.7%	
History of Induced Abortion					
Yes	78	69.6%	84	62.2%	0.22
No	34	30.4%	51	37.8%	
History of Live Birth					
Yes	15	13.4%	43	31.9%	<0.01
No	97	86.6%	92	68.1%	
Household Income (CNY/month)					
≤10,000	38	33.9%	54	40.0%	0.33
>10,000	74	66.1%	81	60.0%	

BMI: body mass index. *p*-values were calculated by univariable regression analysis, and *p* < 0.05 was regarded as a statistical difference.

**Table 3 jcm-11-07474-t003:** Multivariable analysis of risk factors of self-reported depression for RPL patients.

	RR (95% CI)	*p*-Value
Age ≥ 36 years	5.47 (2.42–12.38)	<0.01
Lower education background	1.50 (0.85–2.63)	0.16
>2 times of spontaneous miscarriages	2.94 (1.66–5.22)	<0.01
No history of live birth	3.77 (1.81–7.82)	<0.01

RR: relative risk; CI: confidential intervals. *p*-values were calculated by multivariable regression analysis, and *p* < 0.05 was regarded as a statistical difference.

## Data Availability

The data presented in this study are available on request from the corresponding author. The data are not publicly available due to ethical concerns.

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
