# Peer review of "Self-Reported Depression among Chinese Women with Recurrent Pregnancy Loss: Focusing on Associated Risk Factors"

_jcm, 2022, doi:10.3390/jcm11247474_

Round 1

Reviewer 1 Report

The article submitted for review presents the results of the analysis of the impact of various factors on recurrent pregrancy lost. Please respond to the comments and questions I have after reading the manuscript.

1. The questions contained in the SDS questionnaire are known, however, the authors should include the full custom-made questionnaire in the form of a supplement.

2. Line 103: What was this "previous article"?

3. One of the most important questions I would like to ask: why was the value 53 used as the cutoff value for the SDS scores? Please provide a reference to the literature.

4. Were corrections for the chi square test used? Was only multivariate logistic regression analysis used or was the univariate method also used? If the R environment was used - the exact version of e.g. R studio and the packages used should be provided.

5. Table 1: percentages in groups do not add up to 100%.

6. Why were pregnancy failure questions not asked in the control group?

7. In my opinion, most of the statistically significant results obtained in Figure 1 result from differences in the size of the RPL group and the control group. Depression occurs 15% more often in the study group, so since there are more women aged 35 and over in this group, there will be more cases of depression in this group than in the control group of the same age.
The most important value of the work is the analysis within the RPL group.

8. Figure 1: the total number of women in the RPL group in both age groups is 147 (106+41). Shouldn't it be 247?

9. Table 3: it is stated that the group aged 26 years and over was analyzed. Shouldn't it be 36 years?

10. Figure 2 is illegible and basically unnecessary. The same results can be seen in the table.

11. Each p value in the work should be accompanied by information to which test it refers.

Author Response

Dear Reviewer:

Thank you for your letter and for the reviewers’ comments concerning our manuscript entitled “Self-reported depression among Chinese women with recurrent pregnancy loss: focusing on associated risk factors” (Manuscript ID: jcm-2073358). Those comments are all valuable and very helpful for revising and improving our paper, as well as the important guiding significance to our researches. We have studied comments carefully and have made the revisions according to these comments and recommendations, which we hope meet with approval. Revised portion are marked by using the track changes mode in revised MS document, which was uploaded as a supplementary file and we also uploaded a clean version of the paper. We checked it carefully to ensure that references are within the style of the journal and that title page, authors and affiliations, corresponding author, abstract and keywords are included within the main document.

The main corrections in the paper and the responds to the reviewer’s comments are as flowing:

  1. The questions contained in the SDS questionnaire are known, however, the authors should include the full custom-made questionnaire in the form of a supplement.

Response: Thanks for your kind suggestion. We provided two full custom-made questionnaires in supplementary materials (one questionnaire for women with recurrent pregnancy loss and another for healthy women). In addition, we provided original data of this study as form of a supplement, but only serial number not names of participants were uploaded to protecting the privacy of participants.

  1. Line 103: What was this "previous article"?

Response: Thanks for your comment. We are sorry for the confusing sentence. Before we designed this study, we systematically retrieved previous published studies that explored the association between depression and RPL. All the information collected in this study were determined according to the contents of these studies and our clinical experiences. In the revised manuscript, we corrected the sentence as “All of the information collected above were determined according to previous article exploring the association between depression and RPL, as well as clinical experiences” and cited the references 5, 7, 8, 10.

  1. One of the most important questions I would like to ask: why was the value 53 used as the cutoff value for the SDS scores? Please provide a reference to the literature.

Response: Thank you for your question. There were different cut-off values for SDS scores around the world. In our study, we selected 53 as cut-off value for SDS scores according to previous published cut-off values established basing on 1340 Chinese (reference 12). In reference 12, 42 was calculated as cut-off value of crude SDS scores and 53 was calculated as cut-off of total SDS scores. We provided more detailed description of calculate methods of total SDS scores in “2.3. Questionnaire”.

  1. Were corrections for the chi square test used? Was only multivariate logistic regression analysis used or was the univariate method also used? If the R environment was used - the exact version of e.g. R studio and the packages used should be provided.

Response: Thinks for your comments. We are sorry for the confusing description in the section of “2.4. Statistical Analysis”. During performance of statistical analyses, chi-squared tests were used and if numbers were less than 5 in at least 20% of the cells, Fisher’s Exact Test was performed. Both univariable analysis and multivariable analysis were used. All of the above-mentioned points were provided in the revised manuscript, and we also provided the exact version of R studio and the packages used.

  1. Table 1: percentages in groups do not add up to 100%.

Response: Thinks for your suggestion, we are sorry for our mistakes. We examined the Tables again and corrected the wrong data in Table 1 and Table 2.

  1. Why were pregnancy failure questions not asked in the control group?

Response: Thanks for your question, we are sorry for the incomplete information provided in this manuscript. In this study, 193 healthy women were recruited as controls and were included in control group. All of healthy women in control group have no history of pregnancy before (some were non-pregnant and other pregnant women were in their first pregnancy). Therefore, pregnancy failure questions were not investigated in the control group. The inclusion criteria of control group were improved in the revised manuscript, and we added detailed description in “2.2. Participants” as “One hundred and ninety-three healthy women were selected as controls in this study”. We also supplemented that “Household income (≤10000 or >10000 yuan/month) was investigated only in RPL group because a few non-pregnant women in control group were unmarried” in “2.3. Questionnaires”.

  1. In my opinion, most of the statistically significant results obtained in Figure 1 result from differences in the size of the RPL group and the control group. Depression occurs 15% more often in the study group, so since there are more women aged 35 and over in this group, there will be more cases of depression in this group than in the control group of the same age.

The most important value of the work is the analysis within the RPL group.

Response: Thanks for your comments, we fully agree with your opinion and recognize that many confusing factors may affect the results that “depression occurs 15% more often in the study group”, such as gestational status, age, education background and residence. However, we cannot control the matching of baseline data between two groups because of limited sample size. To explain the influence of difference of baseline data on results, we performed subgroup analyses (as shown in Figure 1). We think the point you mentioned is an important limitation of our study, we added discussion on the limitation in revised manuscript according to your suggestion.

We fully agree with your opinion that the most important value of the work is the analysis within the RPL group. Therefore, we supplement some discussion on the risk factors confirmed in this study, and compared the results of this study with previous studies in the revised manuscript.

  1. Figure 1: the total number of women in the RPL group in both age groups is 147 (106+41). Shouldn't it be 247?

Response: Thanks for your careful review. We are sorry for the mistake. In fact, the number in age ≤ 35 years group for RPL group was “206” as shown in Table 1. We corrected the mistake in revised manuscript.

  1. Table 3: it is stated that the group aged 26 years and over was analyzed. Shouldn't it be 36 years?

Response: Thanks for your careful review. We are sorry for the mistake. We corrected this mistake in Table 3.

  1. Figure 2 is illegible and basically unnecessary. The same results can be seen in the table.

Response: Thanks for your suggestion. We deleted Figure 2 in revised manuscript according to your suggestion.

  1. Each p value in the work should be accompanied by information to which test it refers.

Response: Thanks for your kind suggestion. We added corresponding information according to your suggestion in revised manuscript.

Reviewer 2 Report

The manuscript “Self-reported depression among Chinese women with recurrent pregnancy loss: focusing on associated risk factors” by Rui Gao et al. presents results that a higher proportion of self-reported depression was observed in RPL patients which needs to be addressed. I recommend this paper for publication after substantial minor revision.

1.      The sentence “Depression is a common negative emotion that has been found to be a potential risk factor of RPL.”(line 39-40) is speculative and needs to be more conservative.

2.      The sentence“It is without a doubt that spontaneous pregnancy loss is a traumatic event for women 46 of childbearing age” (line 46-47) is not necessary.

3.      The Participants (line75) did not show whether the chromosomal abnormality associated with RPL is in the fetal compartment.

4.      There were many depression test scales, why choose the Self-Rating Depression Scale (SDS)? The rationale could be expanded (Line 95).

5.      There is no valid discussion of the results. Please further discuss the statistical difference in age, educational background, times of spontaneous miscarriages, and history of live birth in RPL patients, especially the three independent risk factors.

Author Response

Dear Reviewer:

Thank you for your letter and for the reviewers’ comments concerning our manuscript entitled “Self-reported depression among Chinese women with recurrent pregnancy loss: focusing on associated risk factors” (Manuscript ID: jcm-2073358). Those comments are all valuable and very helpful for revising and improving our paper, as well as the important guiding significance to our researches. We have studied comments carefully and have made the revisions according to these comments and recommendations, which we hope meet with approval. Revised portion are marked by using the track changes mode in revised MS document, which was uploaded as a supplementary file and we also uploaded a clean version of the paper. We checked it carefully to ensure that references are within the style of the journal and that title page, authors and affiliations, corresponding author, abstract and keywords are included within the main document.

The main corrections in the paper and the responds to the reviewer’s comments are as flowing:

  1. The sentence “Depression is a common negative emotion that has been found to be a potential risk factor of RPL” (line 39-40) is speculative and needs to be more conservative.

Response: Thanks for your suggestion. We corrected this sentence as “As reported, depression is a common negative emotion that has been found to be associated with RPL” and added corresponding references.

  1. The sentence “It is without a doubt that spontaneous pregnancy loss is a traumatic event for women 46 of childbearing age” (line 46-47) is not necessary.

Response: Thanks for your comments. We deleted this sentence according to your suggestion and corrected the introduction in revised manuscript. We hope the revised manuscript more scientific.

  1. The Participants (line75) did not show whether the chromosomal abnormality associated with RPL is in the fetal compartment.

Response: Thanks for your suggestion. We are sorry that we did not investigate the cause of RPL in this study, as the aim of this study was to explore the self-reporting depression in RPL and healthy group. But we realized chromosomal abnormality both for parents and embryos may be potential risk factors of depression in RPL patients according to your comments, we will include these factors in the future study.

  1. There were many depression-test scales, why choose the Self-Rating Depression Scale (SDS)? The rationale could be expanded (line 95).

Response: Thanks for your suggestion. We explained the reason for selecting the Self-Rating Depression Scale (SDS) in “Introduction” section, as “Self-rating depression scale (SDS) is a simple and reliable tool for assessing depressive symptoms, and they are widely applied in clinical and epidemiological studies [12]. This scale contains 20 items that describe subjective feelings and the manifestation of de-pression, which were all straightaway and self-reported by informant. It is user-friendly therefore suitable for clinicians to decide which patients need more psychological care.”

  1. There is no valid discussion of the results. Please further discuss the statistical difference in age, educational background, times of spontaneous miscarriages, and history of live birth in RPL patients, especially the three independent risk factors.

Response: Thanks for your suggestion. We fully agree with your opinion. We added valid discussion on the results of this study especially on the three independent risk factors, as well as compared the results of this study with previous studies in the revised manuscript.

Round 2

Reviewer 1 Report

The authors applied most of the comments I had to their manuscript. Nevertheless, still, not all changes are satisfactory in my opinion.

1. Lines 128-131: version 3.4.3 is the RStudio version or the "forest plot" package version? Either way, you need to specify the version for the second, undescribed software.

2. Lines 299-300: you wrote that data is available upon request. In this version, you put the datasheet in supplementary files. This fragment in the work needs to be updated or the datasheet needs to be removed.

Author Response

Dear reviewer:

Thanks for your careful comments concerning the revised manuscript entitled “Self-reported depression among Chinese women with recurrent pregnancy loss: focusing on associated risk factors” (Manuscript ID: jcm-2073358). We are sorry that our corrections were not entirely satisfactory, and thank you for pointing out the problem and giving us the opportunity to make further corrections. According to your new comments, we made the following corrections and response point by point:

  1. Lines 128-131: version 3.4.3 is the RStudio version or the "forest plot" package version? Either way, you need to specify the version for the second, undescribed software.

Response: We are sorry for the confusing description. version 3.4.3 is the RStudio version, and the version of "forestplot" package is 3.1.0. We corrected this sentence as “the forest plot for subgroup analysis was performed by “forestplot” package (version 3.1.0) in RStudio, version 3.4.3.”

  1. Lines 299-300: you wrote that data is available upon request. In this version, you put the datasheet in supplementary files. This fragment in the work needs to be updated or the datasheet needs to be removed.

Response: Thanks for your suggestion. We removed the datasheet that was uploaded as supplementary file.

We would like to express our great appreciation to you for comments on our paper. Looking forward to hearing from you.

Thank you and best regards.

Yours sincerely,

Name: Peng Bai

E-mail: jiamu1999@126.com
